# TFIIIC as a Potential Epigenetic Modulator of Histone Acetylation in Human Stem Cells

**DOI:** 10.3390/ijms24043624

**Published:** 2023-02-11

**Authors:** Marco Vezzoli, Lara Isabel de Llobet Cucalon, Chiara Di Vona, Marco Morselli, Barbara Montanini, Susana de la Luna, Martin Teichmann, Giorgio Dieci, Roberto Ferrari

**Affiliations:** 1Department of Chemistry, Life Sciences and Environmental Sustainability, University of Parma, Parco Area delle Scienze 23/A, 43124 Parma, Italy; 2Vall d’Hebron Institute of Oncology (VHIO), C/Natzaret 115-117, 08035 Barcelona, Spain; 3Genome Biology Program, Center for Genomic Regulation (CRG), Barcelona Institute of Science and Technology (BIST) and Universitat Pompeu Fabra (UPF), 08003 Barcelona, Spain; 4CIBER of Rare Diseases (CIBERER), 08003 Barcelona, Spain; 5Institució Catalana de Recerca i Estudis Avançats (ICREA), 08010 Barcelona, Spain; 6Université de Bordeaux INSERM U1312 (Bordeaux Institute of Oncology) 146, rue Léo Saignat, 33076 Bordeaux, France

**Keywords:** TFIIIC, acetylation, H3K18ac, neurogenesis, H3K27ac, hESCs, p300

## Abstract

Regulation of histone acetylation dictates patterns of gene expression and hence cell identity. Due to their clinical relevance in cancer biology, understanding how human embryonic stem cells (hESCs) regulate their genomic patterns of histone acetylation is critical, but it remains largely to be investigated. Here, we provide evidence that acetylation of histone H3 lysine-18 (H3K18ac) and lysine-27 (H3K27ac) is only partially established by p300 in stem cells, while it represents the main histone acetyltransferase (HAT) for these marks in somatic cells. Our analysis reveals that whereas p300 marginally associated with H3K18ac and H3K27ac in hESCs, it largely overlapped with these histone marks upon differentiation. Interestingly, we show that H3K18ac is found at “stemness” genes enriched in RNA polymerase III transcription factor C (TFIIIC) in hESCs, whilst lacking p300. Moreover, TFIIIC was also found in the vicinity of genes involved in neuronal biology, although devoid of H3K18ac. Our data suggest a more complex pattern of HATs responsible for histone acetylations in hESCs than previously considered, suggesting a putative role for H3K18ac and TFIIIC in regulating “stemness” genes as well as genes associated with neuronal differentiation of hESCs. The results break ground for possible new paradigms for genome acetylation in hESCs that could lead to new avenues for therapeutic intervention in cancer and developmental diseases.

## 1. Introduction

Epigenetic regulation of cell identity is a key issue for the use of human stem cells in regenerative medicine and cancer treatment. To date, two distinct multipotent states are generally considered: naïve human embryonic stem cells (hESCs) and human-induced pluripotent stem cells (iPSCs). Whereas the first are derived from the inner cell mass of the blastocyst, the latter offer advantages since they are obtained by somatic cell reprogramming, though they present clinical problems due to their partially differentiated phenotype [1]. Regulation of the histone acetyltransferase (HAT) p300 and its relative CREB binding protein (CBP) are crucial in controlling cellular identity [2,3], and might also be involved in determining the naïve properties of hESCs. During normal growth conditions, p300/CBP have been identified as the major in vivo HATs for histone H3 lysine-18 and lysine-27 acetylation (H3K18ac and H3K27ac) in mouse embryonic fibroblasts (MEFs) [4]. In addition, human lung fibroblasts infected with adenovirus expressing the E1A oncoprotein, whose major target is the p300/CBP complex, displayed global specific hypoacetylation of H3K18 and H3K27 [2,5]. However, no studies have yet been conducted to unveil whether p300/CBP represent the only or most important HATs for H3K18 and H3K27 acetylation in embryonic stem cells. This question is of particular interest due to the significant increase in p300/CBP HAT activity during myogenic cell differentiation [6], suggesting that naïve hESCs might express “stemness” factors that limit p300/CBP activity, which may be subsequently lost upon differentiation. Given this premise, the question to be asked is whether HATs other than p300/CBP could play a role in swaying the landscape of H3K18ac and H3K27ac. The identification of new HAT activities could be a difficult task, especially in pluripotent stem cells. However, recent reports have shown that GTF3C1, the largest subunit of the RNA polymerase III (Pol III) transcription factor C (TFIIIC), directly acetylates H3K18 [7,8,9]. TFIIIC was found to occupy Alu elements (AEs) within the promoters of RNA polymerase II (Pol II) transcribed genes in response to serum starvation, and this was accompanied by increased H3K18ac at the TFIIIC-bound AEs [9]. Curiously, the two most commonly used brands for hESC and iPSC cultures are mTeSR and E8 media [10], both of which are serum-free media. Therefore, it is tempting to propose that p300/CBP might play a smaller role for H3K18ac and H3K27ac in hESCs than previously thought and that TFIIIC could, at least in part, replace p300/CBP as the hESCs HAT for H3K18ac. By analyzing published datasets for genome-wide distribution of H3K18ac, H3K27ac, p300 and TFIIIC in hESCs compared to those in IMR90 human fibroblasts and hESCs-derived neuroectodermal cells, we show that p300 colocalization with H3K18ac and H3K27ac increases with the differentiation of hESCs to fibroblasts and neural lineage. We further unveil that TFIIIC co-localizes with a significant fraction of H3K18ac in hESCs, in the absence of p300. Moreover, our data reveal that TFIIIC occupancy in hESCs is not restricted to pluripotency-associated regions, but extends also to loci linked to neuronal differentiation, indicating that GTF3C1 and its HAT catalytic activity might potentially play a role during hESCs’ differentiation into neurons. These results propose a possible new scenario for genome acetylation in hESCs, which could ultimately lead to new pharmacological roads in cancer therapy and treatment of developmental diseases.

## 2. Results

### 2.1. p300 Occupancy Overlaps with H3K18 and H3K27 Acetylation in Differentiated Cells

In order to better understand the relationship of genome-wide occupancy of p300 and its HAT activity with H3K18ac and H3K27ac during hESCs differentiation, we analyzed published ChIP-seq datasets for these factors [11,12,13,14]. From one study, we compared H3K18ac and p300 ChIP-seq data in H1 hESCs and IMR90 primary fibroblasts, which are fully differentiated embryonic lung fibroblasts (Figure 1A). For each cell line, we ranked all the H3K18ac significant peaks and plotted the p300 signal to the same regions (Figure 1A). This analysis unveiled that the overall level of overlap between p300 and H3K18ac was significantly lower in undifferentiated H1 hESCs than in differentiated IMR90 fibroblasts (~20% versus ~75%, respectively). Therefore, H1 hESCs displayed a larger fraction of H3K18ac that did not correlate with p300 binding (Figure 1B). However, upon differentiation, as for lineage commitment, almost all H3K18 acetylated regions became associated with p300 occupancy (Figure 1A). To test whether the degree of differentiation was linked to the level of co-occupancy between p300 and the H3K27ac mark, we analyzed a dataset of H3K27ac and p300 in H9 hESCs compared to neuroectoderm cells derived therefrom (NEC), which are less differentiated than IMR90 cells, but which are nevertheless fate-committed [14] (Figure 1C). The analysis showed that undifferentiated naïve H9 hESCs exhibited a smaller overlap (~25%) of H3K27ac and p300 binding than the more differentiated NECs (~50%) (Figure 1D). Altogether, this analysis presents evidence suggesting that genome-wide colocalization of H3 lysine-18 and -27 acetylation with p300 increases during cell differentiation. These data also indicate that the fraction of H3K18ac and H3K27ac sites in undifferentiated hESCs lacking p300 binding might be laid out by a different HAT than p300/CBP.

### 2.2. GTF3C1 as a Putative Novel H3K18ac HAT in hESCs

TFIIIC, and in particular its largest subunit GTF3C1, has been shown to acetylate H3K18 in vitro [8] and in vivo upon serum starvation in human T47D luminal breast cancer cells [9]. Given that hESCs are typically cultured in serum-free media [10], and taking into account our previous analyses (Figure 1A–D), we speculated that H3K18ac in hESCs could, at least in part, be deposited by TFIIIC. To provide evidence for this hypothesis, we first analyzed published TFIIIC genome-wide occupancy in H9 hESCs [15] and compared it to H3K18ac and H3K27ac profiles [16].

The results showed that TFIIIC colocalizes with ~1760 H3K18 acetylated regions in H9 hESC, which show very low levels of p300 binding (Figure 2A). These regions were associated with genes important for hESC biology (Figure 2B, GO:Wikypathways), such as *CHD1*, *KLF4*, *HELLS* and many others (Figure 2C), as well as with genes important for neuronal development (Figure 2B, GO:Cellular Component). Importantly, H3K18ac was more represented at these loci than H3K27ac (Figure 2A). In differentiated IMR90, H3K18ac and TFIIIC were significantly lost at these ~1760 loci (Figure 2C), in comparison with a highly expressed fibroblast gene *COL6A3* (Figure 2C). At this locus, H3K18ac and H3K27ac exclusively correlated with p300 and CBP (Figure 2C), in agreement with our previous findings (Figure 1C).

The analysis of the ChIP-seq data supports that TFIIIC may actively participate in H1 naïve hESCs physiology by occupying promoters and enhancers of “stemness” genes (Figure 2C). Surprisingly, TFIIIC is also bound to genes associated with neuronal differentiation (Figure 2B), suggesting that a general transcription factor of the Pol III machinery may potentially play a role in the early stages of neuronal development from naïve hESCs. To support this observation, we employed GIGGLE score analysis [18] to screen ~90,000 available ChIP-seq datasets, from several cell lines and tissues, for transcription factors and chromatin regulators [15,16], and in silico predict the presence of these factors at TFIIIC-bound regions devoid of p300 in H9 hESCs. We found significant enrichment for various neuronal factors such as ADNP, whose mutations underlie a complex neurological disorder (Helsmoortel–Van der Aa syndrome; OMIM#615873), and that interact with TFIIIC in mouse ESCs [19]; CHD4, a chromatin-remodeling enzyme-regulating genome architecture in mouse brain [20], and Polycomb proteins such as RNF2, RYBP, EZH1, EZH2, KDM2B and JARID, all important regulators of neurogenesis and known interactors of TFIIIC in differentiated cells [21,22] (Figure 2D). When we used the same analysis for the TFIIIC-positive H3K18ac-associated regions, we found enrichments for Pol II, the pTEFb-associated protein BRD4, the acetylation-sensitive enhancer regulator BRD2, as well as for many other important factors involved in transcription (Figure 2E). Therefore, these data suggest a possible dual role for TFIIIC in hESCs, actively participating in transcription of stem cell-associated genes, via colocalization with H3K18ac, and occupying neuronal genes before differentiation.

## 3. Discussion

The main goal of this study was to gain further insight into the regulation of histone acetylation and its effects on the identity of hESCs. The results from the analysis of ChIP-seq datasets suggest that p300/CBP could not be the only HAT responsible for H3K18ac and H3K27ac in hESCs, given the genome-wide absence of p300 from many H3K18ac and H3K27ac sites in hESCs. In fact, the overlap of p300 with H3K18ac and H3K27ac regions was only observed upon cell differentiation: partially detected in lineage-committed neuroectodermal stem cells and clearly visible in fully differentiated IMR90 cells. The fact that TFIIIC, and thereby its HAT-containing GTF3C1 subunit [8,9] maps to many H3K18ac sites lacking p300 binding in hESCs led us to propose that TFIIIC could possibly exert a function in H3K18 acetylation in this cell type. We are aware that the results shown here are correlative, but we think that they are suggestive of this possible functional connection. In this context, other enzymes responsible of H3K18 acetylation yet unknown could also substitute p300 in this activity. In addition, we would like to point out that our work focuses on two H3 acetylation marks, and therefore we cannot rule out the requirement of p300 in hESCs for the acetylation of other histones such as H4 which was shown in male germ cells [23].

TFIIIC is not only required for transcription by Pol III, as it has also been shown to play a role in other cellular processes often associated with the segregation of transcriptionally active and repressed domains, and thus with genome and chromatin organization [9,11,20,21]. These activities together with the TFIIIC ability to regulate H3K18ac support a plausible TFIIIC regulatory role for the expression of genes necessary for the differentiation of H9 hESCs into the neural lineage, which deserves further experimental investigation in the future. In fact, a role for TFIIIC in neuron biology was described previously through the regulation of the transcription of synaptic activity-dependent genes [22,23]. The results of our analysis show TFIIIC binding in the absence of H3K18ac at genes involved in many aspects of neuronal differentiation in H9 hESCs, whereas TFIIIC binding overlaps with H3K18ac at stem cell-related genes in H1 hESCs. Though quite speculative at this stage of the work, it is tempting to hypothesize that TFIIIC may play a dual role in stem cell maintenance and differentiation by modulating H3K18ac and/or by acting as a platform for generating protein-protein interactions with other transcriptional machineries, thereby directing complex developmental programs. Regardless, the presence of TFIIIC at H3K18 and K3K27 acetylated loci and the dynamic behavior of both TFIIIC and the acetylated marks suggest a possible new role for TFIIIC in stem cell maintenance and differentiation. In this regard, genome-scale CRISPR-Cas9-based screens have identified GTF3C1 as an important human pluripotency-specific gene [24,25]. However, further studies will be need to delineate the intriguing activities of this 600-kDa Pol III general transcription factor in these processes, including its role in neurogenesis. Moreover, our data discloses a new perspective on the dogma that p300/CBP are the only HATs for H3K18ac and H3K27ac in hESCs, as these marks are observed in these cells even in the absence of mappable p300/CBP binding. By expanding the list of HATs involved in shaping the epigenome of hESCs, our work lays the groundwork for new opportunities for therapeutic intervention, both in cancer (e.g., cancer stem cells) and in developmental diseases resulting from improper differentiation of hESCs.

## 4. Materials and Methods

### 4.1. External Data Sources

ChIP-seq data for p300, H3K18ac, H3K27ac in H1, H9, IMR90 and NEC cells were taken from GEO: GSE16256, GSE17917, GSE43152, GSE24447. ChIP-seq data for TFIIIC in H9 was from GSE195499, while TFIIIC ChIP-seq data in IMR90 (with or without serum) was from GSE120162.

### 4.2. ChIP-Seq Data Analysis

Analysis of ChIP-seq data was carried out as previously described [26] with minor modifications [27]. Heatmaps and average profiles of ChIP-seq data were generated using the computematrix and plotheatmap functions of deeptools 3.0 [28].

### 4.3. Bedtools

Bed intersection was carried out using bedtools [29] “intersectBed” function with default parameters of 1-bp overlap. Graphic representation of heatmaps has been obtained with R graphic, using R-studio (https://www.rstudio.com, accessed on 29 December 2022).

### 4.4. GIGGLE Analysis

Peaks were used for calculating GIGGLE score by using “all peaks” and “chromatin and transcription factor regulation” options in toolkit analysis of Cistrome Data Browser (http://cistrome.org/db/#/, accessed on 29 December 2022).

### 4.5. Genome Browser Images

For genome browser images, the integrative genome viewer (IGV) tool was used.

### 4.6. Gene Ontology (GO) Analysis

g:Profiler (https://biit.cs.ut.ee/gprofiler, accessed on 29 December 2022) [17] was used with default parameters to detect GO terms enrichments and association of peaks with genes, respectively. Lists of genome coordinates derived from downstream analysis of ChIP-seq data were analyzed with g:profiler using default parameters.

## 5. Conclusions

This work brings new knowledge about the role of TFIIIC as a putative hESCs HAT for H3K18ac and thereby as a regulator of stem cells fate These results have a direct impact into creating new possibilities for clinical intervention in cancer and other developmental diseases.

## Figures and Tables

**Figure 1 ijms-24-03624-f001:**
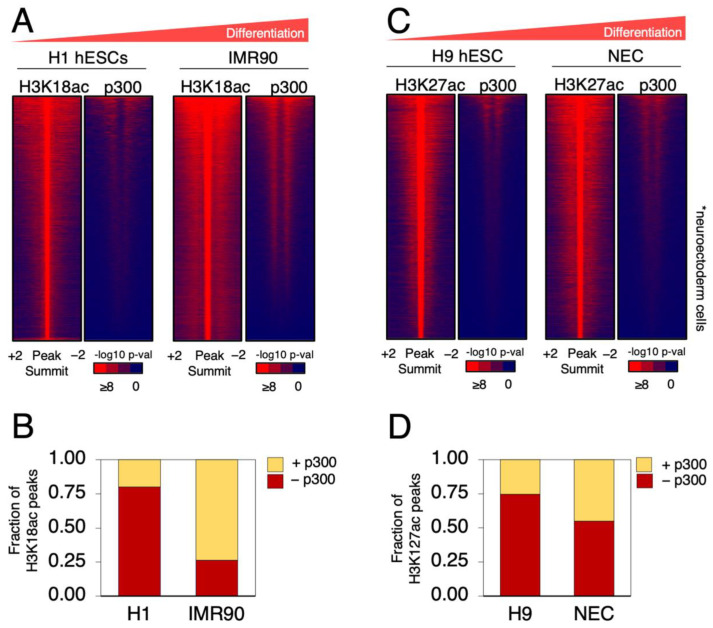
p300 is more associated with H3 acetylated regions in differentiated cells than in H1 and H9 hESCs. (**A**) Heatmap of H1 hESCs H3K18ac and IMR90 H3K18ac sites independently ranked from high to low acetylation. For the same regions of acetylation, the heatmaps of associated levels of p300 in each cell line are also shown. The increase in differentiation is indicated by the red triangle above the figure. Color bar scale with increasing shades of color stands for higher enrichment (−log10 of the Poisson *p*-value). (**B**) Fraction of H3K18ac peaks occupied (+) or not (−) by p300 for the indicated cell lines represented in (**A**). (**C**) Heatmap of H3K27ac peaks in H9 hESCs and NECs shown as in (**A**). The increase in differentiation is indicated by the red triangle above the figure. Color bar scale with increasing shades of color stands for higher enrichment (−log10 of the Poisson *p*-value). (**D**) Fraction of H3K27ac peaks occupied (+) or not (−) by p300 for the indicated cell lines represented in (**C**).

**Figure 2 ijms-24-03624-f002:**
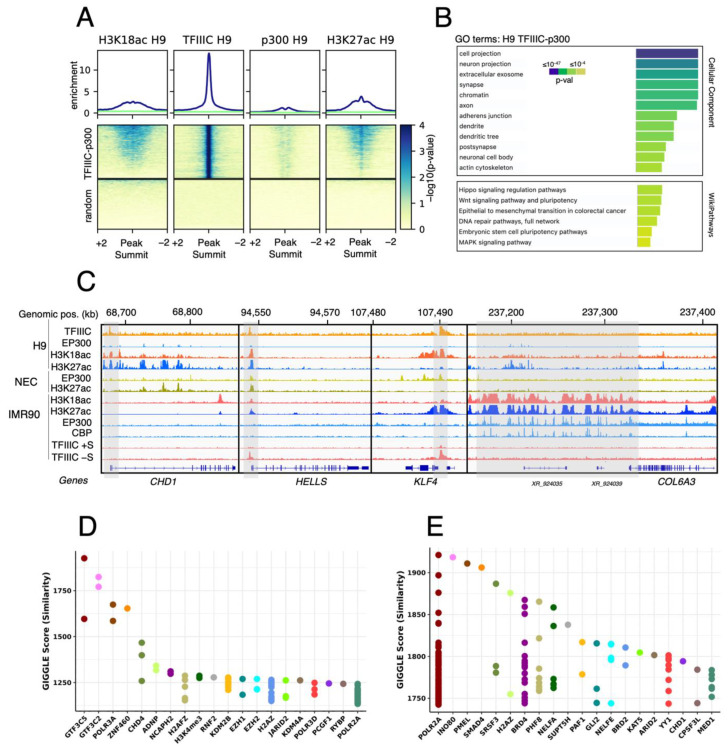
TFIIIC binds close to promoters of stem cells genes and neuronal genes. (**A**) Average levels and heatmap for H3K18ac, TFIIIC, p300 and H3K27ac in H9 hESCs. The panels on top of the heatmaps show the enrichment of the indicated factors at TFIIIC peaks, as seen in the corresponding heatmaps (medium panels). Note that H3K18ac and H3K27ac loci positive for TFIIIC are devoid of p300. The lower panels are heatmaps showing the enrichment of the indicated factors for a random set of peaks. (**B**) g:Profiler analysis [17] of TFIIIC peaks in H9 hESCs, showing GO terms enriched for cellular components and Wikipathways. The *p*-value for enrichment is shown as a color code. (**C**) Genome browser examples of H3K18as and H3K27ac peaks in H9 hESCs, NECs and IMR90 fibroblasts: stem cell-associated genes such as *CHD1*, *HELLS*, *KLF4* and the fibroblast-expressed marker *COL6A3* gene are shown. These stem cell loci are devoid of p300 but associated with TFIIIC in H9 hESCs (the grey bars indicate TFIIIC peaks within the promoter regions of these genes). In contrast, *COL6A3* is associated with p300/CBP in IMR90 cells. At the promoters of *KLF4* and *HELLS*, the TFIIIC peaks are also visible in serum-starved IMR90 cells (–S). (**D**,**E**) GIGGLE score analysis for all TFIIIC peaks (**D**) and TFIIIC peaks overlapping H3K18ac peaks (E) in hESCs [16].

## Data Availability

Data analyzed in this work can be found here: ChIP-seq data for p300, H3K18ac, H3K27ac in H1, H9, IMR90 and NEC cells were taken from GEO: GSE16256, GSE17917, GSE43152, GSE24447. ChIP-seq data for TFIIIC in H9 was from GSE195499, while TFIIIC ChIP-seq data in IMR90 (with or without serum) was from GSE120162.

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
