# Peer review of "TFIIIC as a Potential Epigenetic Modulator of Histone Acetylation in Human Stem Cells"

_ijms, 2023, doi:10.3390/ijms24043624_

Round 1

Reviewer 1 Report

In this manuscript, Vezzoli et al. observed that p300 showed better colocalization with H3K18ac and H3K27ac in differentiated cells than hESCs. Meanwhile, TFIIIC colocalized with ~1760 H3K18 acetylated regions without obvious p300 binding signals in hESCs. In addition, TFIIIC also enriched in neuronal biology-related genes without H3K18ac enrichment in hESCs. Thus, the authors proposed an ambitious model: TFIIIC partially replaced p300 as the hESCs-specific histone acetyltransferase (HAT) for H3K18ac, promoted the “stemness” genes transcription by adding H3K18ac modification and was involved in hESCs differentiation to the neuronal lineage by recruiting neuronal factors.

Overall, the model of this study is of potential interest. However, conclusive evidence is still lacking, and it is an overstatement. The authors didn’t show any data, which can support a causal relationship between TFIIIC and H3K18ac. If the authors want to claim that TFIIIC works as a HAT in hESCs, the changes of H3K18ac must be shown in TFIIIC knockout/knockdown hESCs. Besides, the authors must formally exclude the existence of a protein with HAT activity recruited to the genome by TFIIIC. For TFIIIC’s role in the neuronal differentiation of hESCs, the authors can try neuronal differentiation experiment in TFIIIC knockout/knockdown hESCs. In addition, I didn’t find figure legends for Fig. 2D and 2E.

The quality of this Communication paper does not seem to reach the level of a strong candidate for International Journal of Molecular Sciences. I can’t recommend its publication without significant further data.

Author Response

In this manuscript, Vezzoli et al. observed that p300 showed better colocalization with H3K18ac and H3K27ac in differentiated cells than hESCs. Meanwhile, TFIIIC colocalized with ~1760 H3K18 acetylated regions without obvious p300 binding signals in hESCs. In addition, TFIIIC also enriched in neuronal biology-related genes without H3K18ac enrichment in hESCs. Thus, the authors proposed an ambitious model: TFIIIC partially replaced p300 as the hESCs-specific histone acetyltransferase (HAT) for H3K18ac, promoted the “stemness” genes transcription by adding H3K18ac modification and was involved in hESCs differentiation to the neuronal lineage by recruiting neuronal factors.

Overall, the model of this study is of potential interest. However, conclusive evidence is still lacking, and it is an overstatement. The authors didn’t show any data, which can support a causal relationship between TFIIIC and H3K18ac. If the authors want to claim that TFIIIC works as a HAT in hESCs, the changes of H3K18ac must be shown in TFIIIC knockout/knockdown hESCs. Besides, the authors must formally exclude the existence of a protein with HAT activity recruited to the genome by TFIIIC. For TFIIIC’s role in the neuronal differentiation of hESCs, the authors can try neuronal differentiation experiment in TFIIIC knockout/knockdown hESCs. In addition, I didn’t find figure legends for Fig. 2D and 2E.

The quality of this Communication paper does not seem to reach the level of a strong candidate for International Journal of Molecular Sciences. I can’t recommend its publication without significant further data.

We thank the Reviewer for his/her comments. We are aware that this study shows correlations regarding important events that govern histone acetylation in hESCs and during differentiation. However, the type of experiments and data that the Reviewer requests go beyond the scope of the manuscript prepared as a short communication. We are also aware of the importance of the issue addressed by our study and we think we are offering quite novel perspectives by virtue of correlative genome-wide analysis.

Regarding the Figure Legends, Fig. 2D and 2E were there, but not mentioned in the text. We have amended the Figure legends and add their mention to the text.

Reviewer 2 Report

Vezzoli et al. report on a new regulator of acetylation as TFIIIC in human stem cells. However, the relevance of epigenomic regulation is timely to achieve therapies for cancer and other developmental diseases. This paper used data sets from various external sources to conclude the novel role of TFIIIC in the acetylation process in human stem cells.  

There are major points that will help to make a better impact on this paper.

1.      A discussion pertinent to link the external availability of dietary and nutrient factors as a source of CoA and acetate.

2.      A section on future experimentation is needed.

3.      A separate section TFIIIC and cancer/cancer stem cell will help for better interpretation.

4.      Some in vitro cell-based data on the level of TFIIIC in cancer stem cells or human embryonic stem cells will make better relevance.

5.      The coherence of the paper from the title needs to be restructured so that title should match the strength of the discussion. 

Author Response

RW2:

Vezzoli et al. report on a new regulator of acetylation as TFIIIC in human stem cells. However, the relevance of epigenomic regulation is timely to achieve therapies for cancer and other developmental diseases. This paper used data sets from various external sources to conclude the novel role of TFIIIC in the acetylation process in human stem cells.

We thank the Reviewer for his/her comments, which we have tried to address as shown below.

There are major points that will help to make a better impact on this paper.

  1. A discussion pertinent to link the external availability of dietary and nutrient factors as a source of CoA and acetate.

Although levels of acetyl-CoA are important for histone acetylation in general, we are not fully convinced this may be of importance for this communication. Acetyl-CoA levels will affect all histone acetylation and not only H3K18 and H3K27, as well as many other histone acetylation events.

  1. A section on future experimentation is needed.

We are not aware that this section is a requirement of the Journal. Nevertheless, we have included some comments on the Discussion section on our future interest in this matter.

  1. A separate section TFIIIC and cancer/cancer stem cell will help for better interpretation.

We did add sentences both in the Introduction and Discussion sections about TFIIIC and cancer, and therefore we do not really understand the Reviewer’s request as a section.

  1. Some in vitro cell-based data on the level of TFIIIC in cancer stem cells or human embryonic stem cells will make better relevance.

This is an interesting question, and we therefore looked into expression available data (https://vastdb.crg.eu/) and observed that GTF3C1 levels (as a marker of TFIIIC) are very high in H1 and H9 hESCs as well as in almost all cancer cell lines reported. This indicates the importance of this factor in both hESCs and cancer (see Figure below).

  1. The coherence of the paper from the title needs to be restructured so that the title should match the strength of the discussion.

Based on this suggestion, we have restructured the discussion to better match the title.

Reviewer 3 Report

1-       Authors mentioned in the abstract “  Our analysis reveals that P300 is only marginally associated  with H3K18ac and H3K27ac in hESCs, but mostly overlapped with these histone marks upon differentiation.” My question how are they sure that they say “only” There are some reports that show P300 directly associated with H4- N terminal: https://pubmed.ncbi.nlm.nih.gov/12421817/  and https://pubs.acs.org/doi/10.1021/bi400684q Moreover,  P300 is able to hyper acetylate H4: https://hal.archives-ouvertes.fr/hal-03759791/document Thus the authors should be careful about using “ only”

2-       It would be nice if the authors provide a good graphical abstract to show the pathway of P300 on the acetylation of histone proteins. Perhaps the authors know P300 does not work alone and always controls by Nuclear Protein in Testis (Nut)

3-       In line 197 instead of Materials and Methods it is better to use Methodology since there is no material has been used.

4-       Why the authors studied the activity of P300 in acetylation histone proteins in hEDCs? And what is the difference in epigenetic activity between embryonic and adults?

5-       Authors should compare their results with empirical findings. Perhaps they could result of this paper: https://academic.oup.com/nar/article/39/5/1680/2409413

6-       The authors did not mention the source of the data set that has been used. Moreover, I think the data (processed) has to be out in a public repository.  

7-       I am not surprised that the supplementary is not available in the review panel. ( lines 225-226)

Author Response

  • Authors mentioned in the abstract “Our analysis reveals that P300 is only marginally associated with H3K18ac and H3K27ac in hESCs, but mostly overlapped with these histone marks upon differentiation.” My question how are they sure that they say “only” There are some reports that show P300 directly associated with H4- N-terminal: https://pubmed.ncbi.nlm.nih.gov/12421817/ and https://pubs.acs.org/doi/10.1021/bi400684q. Moreover, P300 is able to hyper acetylate H4: https://hal.archives-ouvertes.fr/hal-03759791/document. Thus the authors should be careful about using “ only”.

We completely agree with this comment, and we have indeed changed the sentence.

  • It would be nice if the authors provide a good graphical abstract to show the pathway of P300 on the acetylation of histone proteins. Perhaps the authors know P300 does not work alone and always controls by Nuclear Protein in Testis (Nut).

We thank the Reviewer for his/her comment. However, the Reviewer is referring to a p300 activity related to H4 acetylation, which we think it is not quite pertinent to this work. Our work focuses on two specific H3 acetylation marks, and therefore the results do not rule out the requirement of p300 for the acetylation of other histones as H4 as has been shown in male germ cells as indicated by the Reviewer. A sentence regarding this question has been added to the Discussion.

3- In line 197 instead of Materials and Methods it is better to use Methodology since there is no material has been used.

We will change “Materials and Methods” to “Methodology”, if the Journal allows it.

4- Why the authors studied the activity of P300 in acetylation histone proteins in hEDCs? And what is the difference in epigenetic activity between embryonic and adults?

We suppose the Reviewer refers to hESCs. We have compared the genome-wide profiles of p300 and TFIIIC (GTF3C1) in undifferentiated hESCs and differentiated fibroblasts to prove the point that in undifferentiated cells TFIIIC might play an important role in both maintenance and differentiation of pluripotent stem cells, because p300 is not present at all H3K18 and H3K27 acetylation sites detected in this cell type. Currently, there is no available data regarding the “epigenetic activity” of p300 in adults, assuming that the Reviewer refers to iPSCs generated by reprograming of somatic cells. Nevertheless, this is a very interesting an important question to be answered in the future.

5- Authors should compare their results with empirical findings. Perhaps they could result of this paper: https://academic.oup.com/nar/article/39/5/1680/2409413

We are not quite sure what the reviewer refers to by pointing to data on how acetylation of H4 is important for the folding of nucleosomes. We are not confident of which type of comparisons he/she is asking for.

6- The authors did not mention the source of the data set that has been used. Moreover, I think the data (processed) has to be out in a public repository.

GEO IDs for each of the datasets used in the work are provided in the Mat&Met section. All files used in the manuscript are processed as described in the Mat&Met.

7- I am not surprised that the supplementary is not available in the review panel. ( lines 225-226) 

We apology for generating confusion to the Reviewer. We have eliminated the section as there is not supplementary material for this manuscript.

Round 2

Reviewer 1 Report

I didn't see that the authors added any valuable data to the revised manuscript! I should point out that the current conclusions of this paper go beyond the scope that data can support. At least, the authors should tone down their statements.

Reviewer 2 Report

The authors have addressed some suggestions.

Author Response

Thanks for your comments.

Reviewer 3 Report

The authors provided the correction and I recommend for Publication. 

Author Response

Thanks for your comments.